# Effect of foot reflexology on chronic pain in Parkinson's disease: A randomized controlled trial

Karel Joineau[1]*, Estelle Harroch[2], Mathilde Boussac[1], Margherita Fabbri[1,2], Clémence Leung[2], Fabienne Ory-Magne[1,2], Vanessa Rousseau[2,3], Patrice Peran[1], Christine Brefel-Courbon[1,2], Emeline Descamps[1,4]

1 Toulouse NeuroImaging Center (ToNIC – UMR1214 INSERM/Toulouse University III), Toulouse, France,
2 Department of Clinical Pharmacology and Neurology, University Hospital of Toulouse, Toulouse, France,
3 MeDatAS Unit, CIC, Toulouse, France, 4 CNRS, Toulouse, France

* karel.joineau@inserm.fr

## Abstract

### Objectives

Effectiveness of Foot Reflexology (FR) on the pain intensity in Parkinson's disease (PD) compared with Sham Massage (SM).

### Design

Monocentric, longitudinal, prospective, double-blind, randomized controlled trial. Randomization with a random number generator in the R software. Fixed-sized block randomization of 3 implemented into Clinsight.

### Participants

Idiopathic PD patients with chronic pain (Visual Analogue Scale (VAS)≥4) were recruited from the Toulouse University Hospital between the 14th of April 2021 and the 25th of May 2025.

### Intervention

Four one-hour long FR or SM sessions three weeks apart with the same specialized FR researcher.

### Main outcome measure

Pain intensity change measured by the mean VAS before and after full completed interventions. The difference was compared between group using a Wilcoxon Mann Witney test. Exploratory outcome: brain functional connectivity

**Data availability statement:** The data that support the findings of this study are available in the Supporting Information files: S1 Data. The raw MRI data are only available on reasonable request to the corresponding author as the protocol did not explicitly stated that the data would be made available. Futhermore they are the property of the ToNIC MRI platform. The protocol is available on the journal webpage of this article.

**Funding:** Funding Sources and Conflicts of Interest: This study was funded by the France Parkinson association to C.B. (https://www.franceparkinson.fr/). The sponsor did not play any role in the study design, data collection and analysis, decision to publish and the preparation of the manuscript. The authors declare that there are no conflicts of interest relevant to this work.

**Competing interests:** There is no conflicts of interests relevant to this work. Financial Disclosures for the previous 12 months: KJ has nothing to declare. EH received support for attending meetings and/or travel from NHC. MB has nothing to declare. MF declares honoraria from LVL Medical, BIAL, AbbVie, and Orkyn for speaking and from consultancy work unrelated to this research. CL received support for attending meetings and/or travel from AbbVie, Elivie, Mertz, Ipsen, Medtronic, Orkyn, and NHC unrelated to this research. FOM served on boards for AbbVie, Aguettant, Elivie, and Orphalan and carried out consulting activities for Aguettant, Abbvie, Orkyn, and NHC. She received travel grants from AbbVie, all of which were unrelated to this research. VR has nothing to declare. PP has nothing to declare. CBC received a research grant from France Parkinson and fees for lectures and consultancy work for Aguettant, Orkyn, Zambon, NHC, and AbbVie, all of which were unrelated to this research. ED has nothing to declare.

## Results

30 PD patients were randomized and analyzed. Interventions were delivered as planned for all patients. Clinical variables did not significantly differed between FR and SM groups. Mean VAS decreased by $-12.3$ mm $\pm 15.2$ in FR group (n = 15) and $-17.9$ mm $\pm 29.4$ in SM group (n = 15). Analyses did not reveal any significant difference between the FR and SM groups (*p-value = 0.88*). There are different patterns in connectivity changes in the medial pain system between responders (at least 30% pain reduction) and non-responders to both therapies. There were no adverse events.

## Conclusion

FR is not more effective than SM in relieving chronic pain in PD. The differences in connectivity patterns within the medial pain pathway may underlie the response to tactile stimulation (FR and SM).

## Trial registration

ClinicalTrials.gov NCT04705207.

## Introduction

Parkinson's disease (PD) is the second most common neurodegenerative disorder in the world and is characterized by the loss of dopaminergic neurons in the substantia nigra and accumulation of alpha-synuclein in the brain. In addition to motor symptoms, non-motors symptoms such as neuropsychiatric disorders, sleep issues, neuro-vegetative symptoms and chronic pain are common. Among them, chronic pain (40–80% of PD patients) is disabling [1–3] and has a major bio-psycho-social impact on activities of daily living, altering the quality of life, professional activity, and social ties of PD patients [1–3].

Currently, chronic pain is still difficult to manage: non-steroidal anti-inflammatory drugs (ibuprofen, diclofenac) and paracetamol are the main treatment proposed to PD patients, despite their only short-term efficacy [4]. Moreover, there are only few controlled trials aimed at relieving PD pain. Thus, there is currently no recommended treatment to efficiently relieve PD pain, which remains a major impairment for PD patients [5]. The use of non-pharmacological interventions (NPIs) should be considered in the context of chronic pain treatment. Indeed, chronic pain is a multimodal experience not solely on nociceptive perception but also on subjective, cognitive and psychological aspects that pharmaceutical drugs cannot address. Among NPIs, foot reflexology (FR) is a specialized massage through the application of controlled pressure on specific areas of the feet called reflex zones, which could restore their homeostasis and improves individuals' physical health and well-being [6]. FR has the potential to stimulate the body to release pain-relieving chemicals, trigger the release of endorphins and enkephalins, which help relieving

pain and improving well-being, and modify the subjective pain experience by enhancing pain acceptance [6]. In a meta-analysis of 44 studies, Lee et al. demonstrated that FR induced a significant decrease of stress, fatigue and pain in various diseases (Lee et al., 2011). In particular, FR is widely studied in cancer where a reduction in the intensity of chronic pain has been demonstrated [7]. However, in the literature concerning FR, there is a lack of reproducibility due to poor and/or unclear methodology. Randomized controlled trials are rare due to the difficulty of setting an appropriate control group. In addition, variability in response to FR is never addressed. Identifying the benefits of a specific intervention for patients suffering from chronic pain is an important direction for research and clinical practice, and homogeneity of protocols through detailed guidance, selection of relevant criteria and appropriate sham group are necessary to validate therapeutic effects [8].

The pathophysiology of pain in Parkinson's disease (PD) involves alterations in pain processing mechanisms, as evidenced by studies showing lower pain thresholds and increased activation of pain-related brain regions after a nociceptive stimulus [9–11]. Functional magnetic resonance imaging (fMRI) has demonstrated alterations in networks involved in attentional processes and executive control following nociceptive stimulation [12]. Additionally, regional disruptions have been observed in pain-related areas, including the orbitofrontal cortex—a region within the medial prefrontal cortex (mPFC)—in Parkinson's disease (PD) patients with pain compared to those without [13]. Furthermore, compared to healthy controls, PD patients with pain exhibited reduced connectivity between the nucleus accumbens and the hippocampus [13]. Finally, correlations have been reported between pain intensity and connectivity between the nucleus accumbens, the insula and various pain-related brain regions, in PD patients experiencing pain and central pain [14,15].

The main objective of this study was to compare the change of pain intensity in PD patients after 4 sessions of FR, compared to a group of patients receiving 4 sessions of Sham Massage (SM). We hypothesized that FR could decrease pain intensity in PD, that the amelioration of pain would be superior to the SM and that pain relief could be linked to specific changes of brain's connectivity.

The secondary aims were to compare: 1) the percentage of responders between the FR and the SM groups; 2) the changes in various pain parameters assessed through different questionnaires between the groups; 3) the changes in heat nociceptive threshold thermotest before and after FR or SM sessions, as well as before and after the long-term intervention in both FR and SM groups; and 4) the variations in pain acceptance, anxiety and depression between both groups. As an exploratory objective, we aimed to identify biomarkers of the FR effects on brain functional connectivity compared to SM.

## Materials and methods

### Study design

This DOREPAR study (ClinicalTrials.gov Identifier: NCT04705207) was a monocentric, longitudinal, prospective, double-blind, randomized controlled study with two parallel groups. It was approved by the CPP Sud-Ouest et Outre-Mer III Ethical committee (ID-RCB: 2020-A03036-33).

Thirty PD patients were included in the study after giving their written informed consent. Randomisation in two equal groups was established by the Methodology and Data Management department (MeDatAS) from the Clinical Investigation Center (CIC) of Toulouse.

There were five visits three-weeks apart: V1 (baseline) to V5 (S1 Fig). During the four first visits, patients underwent sessions of either FR or SM according to their group of randomization. Interventions took place in the Toulouse Neuro-Imaging Centre (ToNIC lab) in France. Pain intensity, questionnaires and brain functional connectivity were assessed at V1 and V5. The nociceptive threshold was assessed before and after interventions at V1 and V4. The patients took their usual dopaminergic medication for the whole duration of the protocol. Adverse events were collected at each visit. Adherence was not measured.

The random allocation sequence was generated using a random number generator built into the R software (block-rand library). A fixed-size block randomization of 3 was used to ensure balance between groups. No stratification was performed. An excel file containing the randomization list was imported into the randomization module of the Clinsight software. The random allocation sequence was generated by an independent statistician. Participants were enrolled by investigators at the site, and intervention assignment was performed by a study coordinator after connection to Clinsight.

All patients, doctors, researchers and individuals analysing the data were unaware of the allocation group. Only the MeDatAS team and the attending care practitioner (E.D.) knew the type of intervention the patients received. The MeDatAS team unblinded the trial following the completion of the statistical analysis.

The entire protocol can be found in S1 Protocol. We ensure reproducibility with the completion of the Consolidated Standards of Reporting Trails (CONSORT) (S1 Checklist) and the CONSORT Extension Non-Pharmaceutical Trials 2017 checklist (S2 Checklist).

### Participants

Idiopathic PD patients (according to the United Kingdom PD Society's Brain Bank), with 18 years old minimum, without disabling dyskinesias (Movement Disorder Society – Unified Parkinson's Disease Rating Scale part IV: MDS-UPDRS part IV < 2; MDS-UPDRS IV.I < 1; MDS-UPDRS IV.2 < 1) and experiencing chronic pain whatever the pathophysiological mechanism (assessed with the classification of *Marques et al., 2019)* for at least 3 months and with a pain intensity ≥ 4 (Visual Analog Scale (VAS)) were included. The dopaminergic medication as well as antalgic treatment had to remain stable for at least 4 weeks before and throughout the study.

Patients with cognitive impairments (Montreal Cognitive Assessment (MoCA) score < 25), participating in clinical trials potentially interfering with the study's objectives, and refractory to foot massage or who had received a reflexology session in the last 6 months were excluded. Furthermore, exclusion criteria covered individuals with contraindications to foot reflexology, such as a history of phlebitis within the last 3 months, skin lesions, or unresolved fractures on the feet, and contraindications to MRI, including claustrophobia or the presence of metallic elements in their body.

Eligibility criteria were verified by doctors of the Parkinson Expert Centre of the University Hospital of Toulouse. The recruitment phase of this study began on the 14th of April 2021 and ended on the 23rd of February 2023. Data were collected at the ToNIC lab and University Hospital of Toulouse.

This study was carried out in accordance with the Declaration of Helsinki. All the patients gave their written and oral consent. Their rights to privacy were observed throughout the study.

### Interventions

**Description of the interventions.** Both interventions were performed by the same researcher (E.D.), certified in reflexology. The patients received a standardized 60-minutes massage (FR or SM), in the same room, temperature and medical chair. Sessions were structured as follow:

- Pre-session dialogue: The practitioner welcomed the patient and gathered information about the patient's bodily sensations and expectations.

- Practice: The patient was comfortably seated on a massage chair, legs slightly raised and barefoot. The practitioner disinfected the patient's feet and then took them in hand, starting with a few relaxation movements to release tension and allow relaxation. Neutral oil was applied, and depending on the group, either the FR or SM protocol was delivered by the practitioner.

- Post-session dialogue: The therapist collected the patient's sensations and feelings, as well as any painful reflex zones.

**Foot reflexology (FR) protocol.** The specific method employed consisted of a specific, non-aggressive and precise solicitation of the cutaneous mechanoreceptors based on body landmarks of the foot map [16]. The technique

is exclusively manual, practiced with the pulp of the thumb or index finger. This precise and fluid stimulation allows application of pressure without inducing pain. The specific FR protocol used in the study had been designed in collaboration with experienced reflexology practitioners and was identical for all patients and all sessions. This protocol stimulated reflex zones associated with pain, stress and emotional management: the reflexes of spine, brain, diaphragm and hypothalamic-pituitary-adrenal (HPA) axis (S2 Fig).

**Sham massage (SM) protocol.** The SM protocol consisted of gentle touch of the foot, starting from the toes to the heel, moving up to the calf, back and circling to the ankle and moving up to the toes. No specific points were pressured, as opposed to FR (S2 Fig).

## Outcome measures

**Primary outcome.** The pain intensity level was assessed before and after interventions using the mean VAS over the last seven days.

**Secondary outcomes.** Other pain parameters were assessed such as the maximal VAS over the last seven days, the Short-Form McGill Pain Questionnaire (SF-MPQ), the Brief Pain Inventory (BPI) and the King's Parkinson Pain Scale (KPPS).

The subjective heat pain perception threshold was assessed before and after sessions at V1 and V4, through repeated thermal stimulations on the skin using a thermotest device (MSA Thermotest, Somedic AB, Sweden) [17]. A contact thermode of 12x25mm was placed on the patient's thenar eminence on the more affected side of the disease (identified through patient interviews and clinical examination with the MDS-UPDRS III).

We used the method of levels which involved starting with an initial thermode temperature of 30°C. Then, for 30 seconds, the patient received a temperature increased by +3°C. Immediately after each thermal stimulation, the patient indicated whether or not they felt a painful sensation (yes/no). In case of a negative response, the temperature was again increased by 3°C and applied for 30 seconds, repeated until the patient perceived pain. In case of a positive response, the temperature was decreased by 1.5°C (half of 3°C) and applied again for 30 seconds. The threshold was then gradually adjusted until a temperature was reached between two intervals close to 0.2°C. A 30-seconds interval was maintained between each stimulation.

The anxio-depressive state was assessed using the Hospital Anxiety and Depression (HAD), the pain acceptance using the Chronic Pain Acceptance Questionnaire 8 items (CPAQ-8) and the motor state using MDS-UPDRS III at V1 and V5.

**Exploratory outcome.** MRI images were acquired at V1 and V5 and included structural (T1) and resting-state functional (rs-fMRI) imaging on a Philips 3T. For T1 images: MPRAGE sequences were conducted with a 3D sagittal acquisition, square FOV = 240 mm, 1 × 1 × 1 mm3, TR = 7.5ms, TI = 900ms, TE = 3.5ms, flip angle = 8°, no fat suppression, full k-space, no averages. For functional images: EPI sequences were conducted with a nominal voxel size of 3 × 3 × 3 mm3, TR = 2.0s, TE = 30ms, α = 90° (Ernst angle), equidistant interleaved slice order, 3mm slice gap, 41 slices, 300 volumes, and no parallel imaging. Acquisition time was 10 min. Participants were instructed to relax, keep their eyes closed and try to avoid thinking about anything at all.

Data were processed with the CONN toolbox version 21.a (http://www.nitrc.org/projects/conn) [18].

First, functional data were realigned; all scans were co-registered and resampled to a reference image. Then, temporal misalignments between different slices were corrected. Outlier scans were identified based on the global Blood Oxygen Level Dependant (BOLD) signal and subject motion in the scanner, and these outlier scans were excluded from the dataset. Following this, both functional and anatomical data were normalized into standard Montreal Neurological Institute (MNI) spaces and segmented into grey matter, white matter, and Cerebral Spinal Fluid (CSF) tissue classes. The functional data underwent smoothing through spatial convolution using a Gaussian kernel with a full-width half-maximum of 8 mm [19]. After smoothing, potential confounding effects on the estimated BOLD signal were estimated and removed separately for each voxel, subject, and functional run using the aCompCor method [20]. This method included noise

components from cerebral white matter and cerebrospinal areas, estimated subject-motion parameters, identified outlier scans or scrubbing, and constant and first-order linear session effects. Finally, the residual BOLD time series underwent band-pass filtering within a low-frequency window of interest (0.009 Hz < f < 0.08 Hz). Quality control plots were analysed to assess the denoising process before proceeding to the first-level analysis.

We have chosen to explore the Seed-to-Voxel functional connectivity of three cerebral regions crucial for chronic pain and parkinsonian chronic pain as seen in the introduction: the insula, the accumbens, and the mPFC [12–14,21,22].

The anterior and posterior parts of both left and right insulae were separated into 10 mm radius spheres (MNI: right posterior insula +39–15 + 08, left posterior insula −39–15 + 01, right anterior insula +32 + 16 + 6, left anterior insula −32 + 16 + 6) [23]. Coordinates established by Baliki et al. (2012) were used for the mPFC (+2, + 52, −2) with a 10 mm radius sphere. The Harvard-Oxford atlas, directly implemented in CONN, was used to extract the left and right nuclei accumbens masks (−10 + 8–10 and +10 + 12–8).

In the first-level analyses, Pearson's coefficients were calculated between the seed's time course and the time courses of all other voxels in the brain, and then transformed into normally distributed scores using Fischer transformation. Then, a general linear model was computed for testing statistical hypotheses in the second-level analyses. For clustering, the Gaussian Random Field theory method was applied with a cluster threshold corrected for multiple comparisons (p < 0.05).

In ROI-to-ROI, we have chosen to explore the functional connectivity in several network implicated in the sensory and attentional processes since they are altered in PD patients after a nociceptive stimulation [12]: the Default Mode Network (DMN), the Executive Control Network (ECN), and the Salience Network (SN) [24,25]. The Sensory Motor Network (SMN) is also implicated in attentional processes, involved in tactile touch and pain [24–26]. We also explored the newly formed Neural Network Correlates of Pain (NNCP) from Descamps et al., 2023 including 23 regions in relation with pain and chronic pain. For this network, we used the Harvard-Oxford atlas implemented in CONN.

A representative BOLD time course was derived for each Region of Interest (ROI) by averaging the signals from all voxels within that ROI. Pearson's coefficients were computed between the time courses of each pair of ROIs, and transformed into normally distributed scores using Fischer's transformation. Then, a general linear model was applied for the second-level analysis. For clustering, we employed the functional network connectivity method outlined by [27]), using multivariate statistics to explore differences in connectivity across our groups and visits. A cluster threshold (p < 0.05) corrected for multiple comparisons was used.

## Statistical analyses

With an expected mean difference between FR and SM groups of 20 mm on the mean VAS and a standard deviation of 15, twelve evaluable patients per group were required (two-sided significance level at 5% and power of 90%). Presuming a drop-out rate of 25%, fifteen participants per group were planned to be randomized with a 1:1 allocation rate.

As descriptive analyses, mean and standard deviation were calculated for quantitative variables, and headcount and percentage for qualitative variables. The clinical and connectivity variables were compared between both groups of interventions (FR vs MS) using Wilcoxon-Mann-Whitney tests. For the gender and the origin of pain, chi2 tests were used.

For the main objective, the change of the mean VAS between before and after intervention was calculated and compared between groups using Wilcoxon-Mann-Whitney tests.

For the secondary objectives, we compared the percentage of responders (pain decrease between V1 and V5 superior or equal to 30%) between groups of interventions using a chi2 test.

The variations between V1 and V5 of the other pain measures as well as the CPAQ-8, HAD and MDS-UPDRS III were also compared between groups with Wilcoxon-Mann-Whitney tests.

For the nociceptive threshold, the changes between before and immediately after the intervention were calculated for each visit (V1 and V4) and compared between groups using Wilcoxon-Mann-Whitney test. Then, changes between before

and after full protocol were calculated for each condition (before and immediately after the intervention) and also compared between groups.

For every clinical variable (except the nociceptive threshold), intragroup changes were assessed using signed-ranks Wilcoxon tests.

For ROI-to-ROI and Seed-to-Voxel analyses, we performed 2x2 repeated measure ANOVAs with groups of interventions (FR and SM) and visits (before (V1) and after (V5) intervention) as factors on functional connectivity measures. The changes in connectivity were then correlated with the VAS changes using Spearman tests. We also performed supplementary ROI-to-ROI and Seed-to-Voxel analyses: we investigated the effects of both tactile foot stimulation on connectivity, independently of the interventions, with a t-test on connectivity values between V1 and V5; and we compared the difference in connectivity between responders and non-responders (defined as a decrease ≥ 30% in mean VAS for responders) with a 2x2 repeated measures ANOVA.

Statistical analyses on clinical variables were performed on Jamovi (version 2.4.1).

All statistical analyses involving connectivity were performed directly on CONN under Matlab (R2021b). Data of significant clusters were then exported in Jamovi to perform the post-hoc tests with Bonferroni corrections for multiple comparisons. Figs were realized with RStudio (2023.12.1) and CONN.

## Results

### PD Patients

The intervention phase began with the first patient first visit (FPFV) on 14th of April 2021 and continued until the last patient last visit (LPLV) on the 25th of May 2023, which marks the final scheduled visit of the last enrolled participant. Thirty PD patients with chronic pain were included and completed this study (Fig 1). One patient in the SM group did not undergo the MRI sessions because of fear during the exam. The first intervention was administered one hour after the randomization. There were no drop-out. The study ended after completion. No adverse event was reported. The interventions were realized as planned.

Patients had a mean age of 65.8 ± 7.4 years old, suffered from PD for 7.4 ± 5.2 years, and from pain for 9.0 ± 8.7 years, with a mean VAS of 56.0 ± 14.3 and a maximal VAS of 73.7 ± 16.0 at baseline.

There was no significant difference between the two groups of patients at V1 on demographic and clinical data (Table 1).

### Primary outcome

Mean VAS change from V1 to V5, decreased by −12.3 mm ± 15.2 in FR group and −17.9 mm ± 29.4 in SM group. Analyses did not reveal any significant difference between the FR and SM groups (*p-value = 0.88*, Table 2).

### Secondary outcomes

There was no significant difference in the percentage of responders (40% in the FR group and 53% in the SM group, *p = 0.46*).

Concerning the other pain measures, the anxio-depressive and motor states, only significant differences were found for the KPPS change (+0.47 ± 14.6 in the FR group vs −12.5 ± 13.8 in the SM group, p = 0.02) and the CPAQ-8 change (+3.47 ± 2.97 in the FR group vs −2.13 ± 8.59 in the SM group, *p = 0.03*) (Table 2).

The nociceptive threshold did not show any significant differences between the two groups neither short- nor long-term effect (*p > 0.05*).

In the FR group, we found significant reduction in the mean VAS (*p = 0.007*), and significant improvement in CPAQ-8 willingness (*p = 0.007*) and CPAQ-8 total (p = 0.001). In the SM group, there was a significant decrease in the KPPS score (*p = 0.008*) (Table 2).

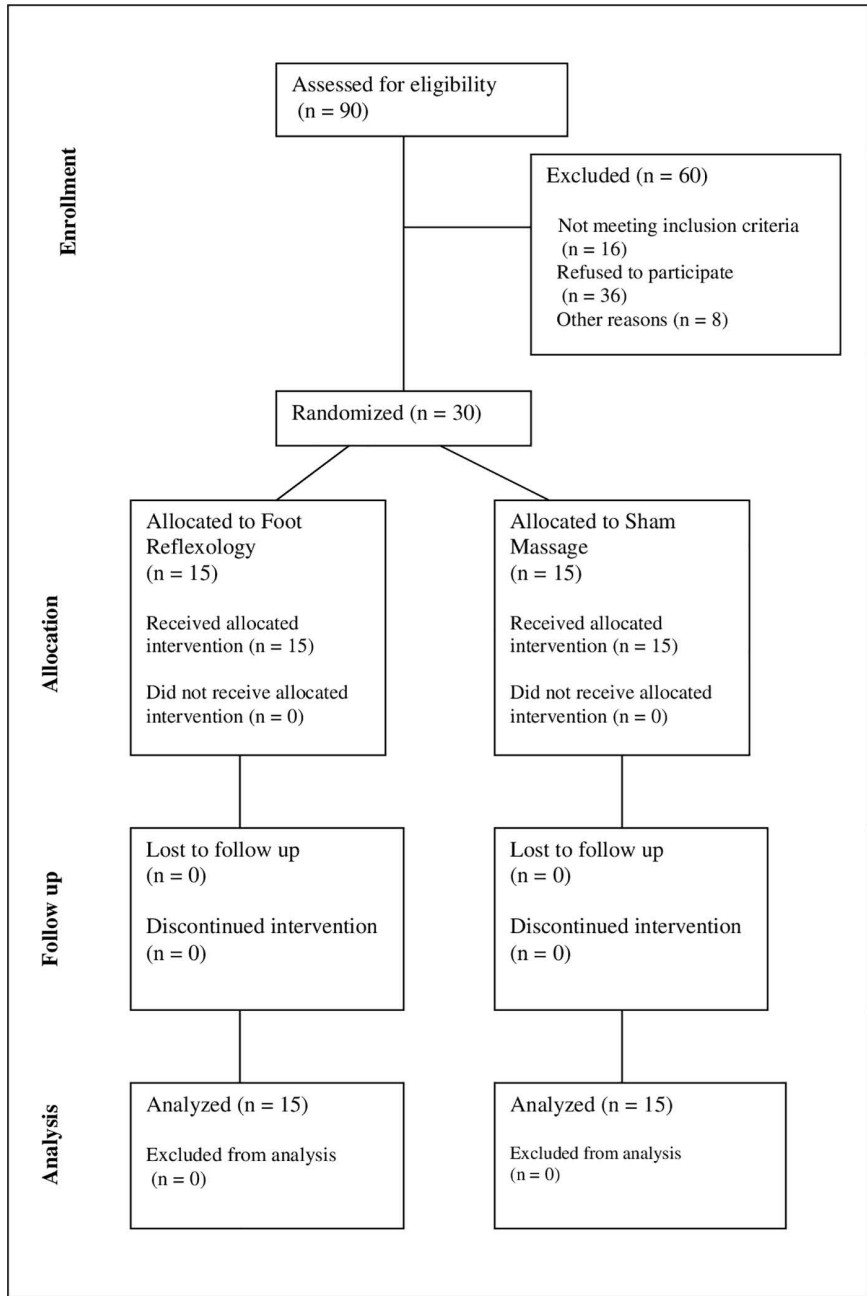

**Fig 1. CONSORT diagram showing the flow of participants through each stage of the DOREPAR randomized trial for the main outcome (change of VAS).**

## Exploratory objective

For this analysis, there were 15 FR patients and 14 SM. The 2x2 mixed ANOVA did not show any interaction between factors, groups and visits, in ROI-to-ROI analyses for any networks (NNCP, DMN, SMN, ECN and SN). In Seed-to-Voxel analyses, group-specific changes in functional connectivity (FR vs SM) between the left accumbens and two clusters

**Table 1. Demographic and clinical data at baseline.**

| | PD patients (n = 30) | | | FR Group (n = 15) | | | SM Group (n = 15) | | | Difference between groups p-value |
|---|---|---|---|---|---|---|---|---|---|---|
| Gender (women) | 18 (60%) | | | 7 (47%) | | | 11 (73%) | | | 0.14 |
| Age (years) | 65.8 | ± | 7.41 | 64.7 | ± | 8.00 | 67.0 | ± | 6.86 | 0.55 |
| PD duration (years) | 7.37 | ± | 5.22 | 6.87 | ± | 4.14 | 7.87 | ± | 6.22 | 0.93 |
| Pain duration (years) | 9.03 | ± | 8.69 | 9.00 | ± | 9.78 | 9.07 | ± | 7.80 | 0.69 |
| LEDD (mg/day) | 721 | ± | 423 | 681 | ± | 417 | 761 | ± | 440 | 0.63 |
| MDS-UPDRS III | 25.1 | ± | 15.8 | 27.3 | ± | 14.1 | 22.9 | ± | 17.5 | 0.25 |
| mean VAS | 56.0 | ± | 14.3 | 55.1 | ± | 13.5 | 56.9 | ± | 15.5 | 0.74 |
| max VAS | 73.7 | ± | 16.0 | 72.4 | ± | 14.2 | 75.1 | ± | 18.0 | 0.77 |
| KPPS | 24.5 | ± | 17,3 | 19.7 | ± | 10.1 | 29,2 | ± | 21,7 | 0.28 |
| BPI | 30.3 | ± | 11.7 | 28.5 | ± | 12.5 | 32.1 | ± | 10.9 | 0.33 |
| SF-MPQ total | 16.7 | ± | 8.30 | 17.7 | ± | 8.35 | 15.7 | ± | 8.42 | 0.48 |
| SF-MPQ sensory | 12.1 | ± | 5.95 | 13.3 | ± | 6.04 | 10.9 | ± | 5.80 | 0.28 |
| SF-MPQ affective | 4.60 | ± | 3.16 | 4.40 | ± | 2.82 | 4.80 | ± | 3.55 | 0.88 |
| CPAQ-8 total | 26.7 | ± | 7.63 | 24.6 | ± | 7.20 | 28.9 | ± | 7.68 | 0.17 |
| CPAQ-8 willingness | 9.93 | ± | 4.99 | 8.67 | ± | 4.42 | 11.2 | ± | 5.35 | 0.20 |
| CPAQ-8 activity engagement | 16.8 | ± | 5.03 | 15.9 | ± | 5.43 | 17.7 | ± | 4.61 | 0.50 |
| HAD total | 16.0 | ± | 6.11 | 16.9 | ± | 5.59 | 15.1 | ± | 6.72 | 0.39 |
| HAD Anxiety | 9.34 | ± | 3.82 | 9.87 | ± | 4.02 | 8.79 | ± | 3.66 | 0.55 |
| HAD Depression | 6.69 | ± | 3.63 | 7.00 | ± | 3.72 | 6.36 | ± | 3.63 | 0.79 |
| Heat nociceptive threshold (°C) | 41.8 | ± | 2.49 | 41.1 | ± | 2.54 | 42.4 | ± | 2.34 | 0.19 |
| Types of pain | | | | | | | | | | 0.64 |
| Musculoskeletal | 9 (30%) | | | 4 (26%) | | | 5 (33%) | | | |
| Radicular | 1 (3.3%) | | | 0 | | | 1 (7%) | | | |
| Restless Leg Syndrome | 1 (3.3%) | | | 1 (7%) | | | 0 | | | |
| Dystonic | 1 (3.3%) | | | 1 (7%) | | | 0 | | | |
| Central | 13 (43.3%) | | | 6 (40%) | | | 7 (46%) | | | |
| Other | 5 (16.7%) | | | 3 (20%) | | | 2 (13%) | | | |

Means ± standard deviations; MoCA: Montreal Cognitive Assessment; LEDD: levodopa equivalent daily dose; MDS-UPDRS III: Movement Disorder Society – Unified Parkinson's Disease Rating Scale part III; VAS: Visual Analogue Scale; KPPS: King's Parkinson Pain Scale; BPI: Brief Pain Inventory; SF-MPQ: Short Form McGill Pain Questionnaire; CPAQ-8: Chronic Pain Acceptance Questionnaire 8; HAD: Hospital Anxiety and Depression.

(subcallosal cortex and right accumbens) were found (S3 Fig), but they did not correlate with the mean VAS change (p = 0.88; p = 0.56).

## Supplementary analysis

As these parameters were not correlated, we investigated the effects of both tactile foot stimulation on connectivity, independently of the interventions, on connectivity values between V1 and V5. In ROI-to-ROI analyses, the SMN did show a significant decrease in connectivity between V1 and V5 for a cluster formed of the left pre- and post-central gyri, the right pre- and post-central gyri and the supplementary motor area (S4 Fig) but this connectivity changes did not correlate with the mean VAS change (p-value = 0.79). No other change was found in any of the DMN, ECN, SN and NNCP networks.

Second, we compared the difference in connectivity between responders and non-responders. As shown in S1 Table, there was no significant difference in clinical variables between the responders and non-responders. In connectivity analysis, there were 14 responders and 15 non-responders. At baseline, the connectivity between responders and

**Table 2. Changes in clinical variables.**

| Changes between V1 and V5 | FR Group | | | SM Group | | | Difference between groups p-value |
|---|---|---|---|---|---|---|---|
| Mean VAS | −12.3 | ± | 15.2* | −17.9 | ± | 29.4 | 0.88 |
| Maximal VAS | −11.3 | ± | 20.9 | −15.7 | ± | 36.1 | 0.89 |
| KPPS | 0.5 | ± | 14.6 | −12.2 | ± | 13.8* | **0.02** |
| BPI | −4.3 | ± | 10.9 | −5.5 | ± | 11.9 | 0.75 |
| SF-MPQ total | −0,2 | ± | 6.4 | −2.1 | ± | 6.4 | 0.41 |
| SF-MPQ sensory | −0.3 | ± | 4.9 | −1.3 | ± | 5 | 0.75 |
| SF-MPQ affective | 0.1 | ± | 2.5 | −0.8 | ± | 2.8 | 0.57 |
| CPAQ-8 total | 3.5 | ± | 3.0* | −2.1 | ± | 8.6 | **0.02** |
| CPAQ-8 willingness | 2.5 | ± | 2.8 | −1.1 | ± | 7.1 | 0.08 |
| CPAQ-8 activity engagement | 0.9 | ± | 3.7 | −1.1 | ± | 4.1 | 0.17 |
| HAD total | −1.5 | ± | 3.4 | 1.3 | ± | 5.6 | 0.15 |
| HAD Anxiety | −0.7 | ± | 2.3 | −0.1 | ± | 2.9 | 0.49 |
| HAD Depression | −0.7 | ± | 2.5 | 0.5 | ± | 3 | 0.25 |
| MDS-UPDRS III | −1.1 | ± | 8.4 | −1 | ± | 6.7 | 0.86 |

Means ± standard deviations; VAS: Visual Analogue Scale; KPPS: King's Parkinson Pain Scale; BPI: Brief Pain Inventory; SF-MPQ: Short Form McGill Pain Questionnaire; CPAQ-8: Chronic Pain Acceptance Questionnaire 8; HAD: Hospital Anxiety and Depression; MDS-UPDRS III: Movement Disorder Society – Unified Parkinson's Disease Rating Scale part III. Intergroup changes differences assessed with Wilcoxon-Mann-Witney tests and

*Significant differences between V1 and V5 in each group assessed with signed-ranks Wilcoxon tests (intragroup differences between V1 and V5).

non-responders showed a significant difference in connectivity between the mPFC and a cluster in the left thalamus (+0.12, ± 0.03 *(SEM), p = 0.004*). After intervention, there was a significant decrease of this connectivity in the responders' group (mean at V1: 0.032 ± 0.075, mean at V5: −0.094 ± 0.095, difference: −0.126 ± 0.082*, p < 0.001*) and a significant increase in the non-responders' group (mean at V1: −0.092 ± 0.093, mean at V5: 0.026 ± 0.070, difference: 0.118 ± 0.090, *p < 0.001*) (Fig 2), which correlates with changes in mean VAS (*Rho = 0.789, p < 0.001*).

## Discussion

In this study, we report that FR does not reduce significantly more pain intensity than SM in PD patients. Moreover, the percentage of responders were similar in the two groups. These two results suggest that FR has no specific analgesic effect on chronic pain in PD compared with SM.

The specific effect of FR has never been studied in PD. According to recent reviews [28–30], the effectiveness of FR over SM was only demonstrated in acute pain.; while in chronic pain, the methodological quality of studies is poor, and the results are discordant. The efficacity of FR was only demonstrated compared with standard care in various pain conditions (rheumatoid arthritis, multiple sclerosis, and migraine). Compared with SM, the results were inconsistent with only one study out of three concluding to a superior effect (cancer chronic pain). This tends to support our finding.

Nevertheless, we observed a reduction in several pain parameters in both groups. Consequently, SM appears to be as effective as FR in reducing chronic pain in PD patients. In this idea, recent reviews of RCTs confirmed the efficiency of manual therapies (consisted of either body massage, FR or feet massage similar to the one used in our control group) to relieve pain in various chronic pain conditions such as tension headaches, low back pain and neck pain [31–33]. In PD, three papers have reviewed the effect of manual therapies and concluded to a moderate benefit in pain improvement [34–36], supporting our results. Manual therapies varied from soft to hard massage, but control group was frequently lacking. After a single FR or SM session, an increase in well-being in healthy participants has been reported [16], which could explain our results in PD patients by modulating the emotional aspects of pain [37].

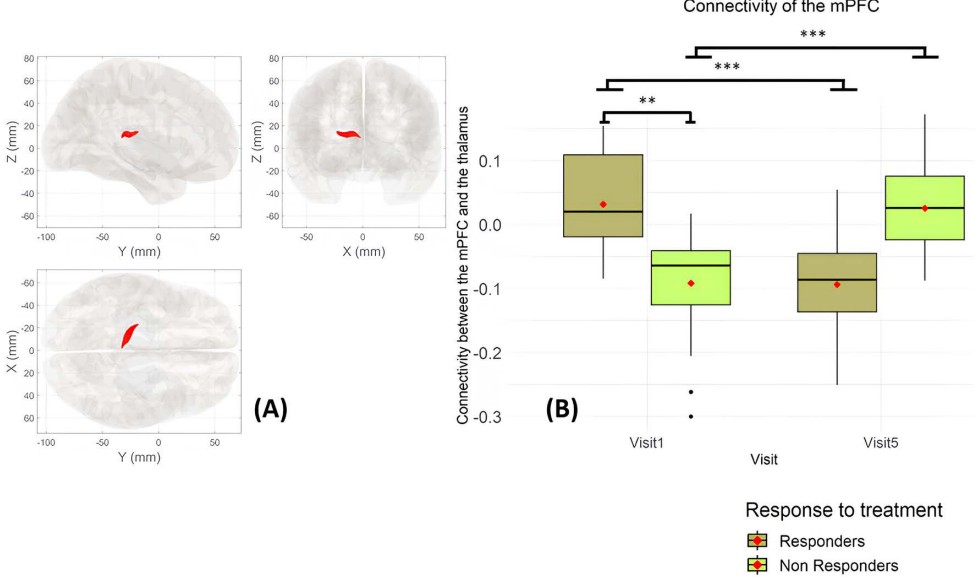

**Fig 2. Changes in connectivity of the mPFC among responders and non-responders.** A) 3D representation of the cluster −16–26 + 12 formed of 168 voxels with 97 voxels covering 7% of the left thalamus and 71% covering a non-labelled area. *size p-FDR = 0.011661*, size *p-unc = 0.001014*. B) Boxplots of the connectivity between the mPFC and this cluster in responders and non-responders and at each visit. Dark green = responders; green = non-responders.

The reduction of pain in both groups could also be attributed to a placebo effect. In placebo analgesia, two mechanisms are critical: expectations (positive or negative) and learning [38]. Positive expectations could lead to a reduction in the affective aspect of pain and activate reward mechanisms after pain relief [38]. Here, talking and listening before and after the one-hour sessions could have enhanced positive expectations by increasing the patient feeling that he is taking care of. Indeed, the relationship between the practitioner and the patient is primordial in the placebo effect [39]. Also, the repeated rituals could also have instigated a Pavlov's conditioned reflex into feeling better [40] by remembering previous pain relief experiences and the reward that followed.

We cannot exclude that the inclusion of a larger number of patients could have highlighted a difference in pain reduction between the two groups. In addition, the high variability of pain response in the SM group could partly explain the lack of significance. Despite the possible placebo effect that could demean the interpretations of this study, it would be interesting to investigate in what extent the context and rituals of a NPI could be maximised in clinical practice to increase its benefits.

Regarding the pain acceptance, the specific increase in the CPAQ-8 solely in the FR group was too small to conclude to a real clinical amelioration.

There was no specific change of the heat nociceptive threshold after one session and after the full protocol in both groups meaning that FR did not modify the abnormal pain processing of PD patients. Despite recent findings concluding to a probable amelioration of sleep, anxiety and depression by FR [33], we did not show any specific effect of FR (or tactile touch) on the anxio-depressive state. There were no adverse events in the two groups during this study, so we can validate these two interventions as safe, in accordance with Huissoud et al. 2023.

Concerning imaging analyses, there were distinct connectivity changes in the limbic system between the nucleus accumbens, subcallosal cortex and frontal medial cortex between the FR and SM groups but these changes were sparse and unrelated to pain intensity despite the role of the limbic system in the emotional aspect of pain [26].

Following both tactile stimulations, we observed a reduction in functional connectivity within the somatosensory and motor regions, coherent with resting state imaging studies showing changes in activation in these regions during or after massage or reflexology sessions [41–43].

However, Descamps et al. found a decrease in connectivity during a FR or SM session in healthy subjects and not an increase as in our results. This difference could be due to the particular nature of our patients and to the temporality, since the MRI was acquired 3 weeks after the end of the sessions. Despite the involvement of these regions in pain processing [26], the changes in connectivity were not linked to pain reduction.

Because none of these connectivity changes correlated with pain reduction, we have chosen to separate patients regarding their response to both interventions to seek differences in their connectivity. In baseline, responders had higher connectivity than non-responders between the mPFC and the thalamus. After interventions, responders showed a decrease in connectivity between these areas while non-responders showed an increase. The thalamus conveys pain information coming from the periphery to cortical regions including the mPFC which modulates the attention and the emotional evaluation of pain through top-down controls [26].

We hypothesize that the increased connectivity in baseline in responders may reflect a more efficient top-down modulation of pain, facilitating treatment response. The decrease of connectivity in responders after treatment could illustrate a reduced compensatory mechanism since the interventions are partly effective. In the opposite, the increase of connectivity in non-responders could illustrate a late compensatory mechanism that would be ineffective. After a FR session, the thalamus was activated [41,43] and its connectivity with other parts of the medial pain system decreased after tactile stimulations in healthy subjects and patients with acute pain [16,44]. This is in agreement with the decrease in connectivity in our responders. In the present study, the different connectivity patterns between the groups could illustrate different underlying mechanisms of analgesic response to tactile foot stimulation within the medial pain system.

To our knowledge, this is the first RCT studying the effects of FR in PD pain compared to SM. The strength of this study is its reproducibility and detailed method using a strong comparative group as control. There were also some limitations. First, the sample size was quite limited and could have led to a lack of statistical power. The duration and frequency of the sessions could also be discussed as these parameters vary considerably from one study to another in the literature: the benefit of NPIs could have been greater with more closely spaced sessions. In addition, we have chosen to include all types of pain because we had no preconceptions on the efficacy of FR on a specific type of pain. The effect of FR or SM should be investigated for each type of pain in the future. Finally, due to the sparsity of antalgic treatments, the dosage could not have been compared between groups or added as covariable.

Lastly, Hohenschurz-Schmidt et al. 2022 [45] advise to design control interventions that are as similar as possible to the tested NPI, even if building a suitable control group that guarantees double-blinding is extremely complex. Indeed, an intervention can be divided into specific and non-specific factors (contextual effects as seen above in the placebo effect such as the patient-practitioner relation, patient's expectations, rituals). In this protocol, SM included all the non-specific elements of reflexology, and was very similar to FR involving the same contextual effects as well as touch on the feet. The only difference was the type of movement and location of stimulated zones position. However, it is often uneasy to see what falls under specific or non-specific factors in NPI. Indeed, elements classified as non-specific in drug trials may be an integral part of a NPI therapy. We could have denatured the FR intervention, which is personalized in current practice, with a standardised intervention that might result in underestimation of the real FR intervention and building an effective SM. Methodology requires further reflection and needs to be continued for continuous improvement.

In conclusion, both FR and SM could be effective in relieving chronic pain in PD. The current challenge may be to determine which patient profiles respond best to foot tactile stimulation.

## Supporting information

**S1 Fig. Design of the protocol.** V: Visit, FR: Foot Reflexology, SM: Sham Massage.
(TIFF)

**S2 Fig. Schema of the reflexology circuit and the stimulated reflex zones. a. Internal arch of left foot view, b. Big toe, plantar arch of left foot view, c. Plantar arch of left foot view.** *A. Base of distal phalange of large toe = C1 vertebra reflex, B. Medial process of calcaneus = coccyx reflex, 1. occipito-cervical junction reflex, 2. brainstem reflex, 3. 4th ventricle and cerebellum reflex, 4. sphenobasilar symphysis, pituitary, hypothalamus reflex, 5. tent of cerebellum reflex, 6. sphenobasilar symphysis reflex, 7. hippocampus reflex, 8. septum pellucidum reflex, 9. corpus callosum reflex, 10. epiphysis reflex, 11. diaphragm reflex, 12. diaphragm insertion reflex, 13. adrenals reflex, 14. epiphysis reflex, 15. hypothalamus reflex, 16. pituitary reflex.* **FR circuit:** 1) Starting with the left foot, then the right foot. Relaxation movement: complete smoothing of foot x3, ankle smoothing x3. 2) Starting with the spine reflex: the pathway followed is from point A to point B (in blue), then from point B to point A (in purple), see S2 Fig a. 3) Starting with the left foot, then the right foot. Then, the 10 reflex zones associated with the brain reflex, as shown in S2 Fig b, zone 1 to zone 10, are successively stimulated on the big toe, with strong stimulation in the upper (locus niger) and middle (raphe nuclei) parts. Next, the beginning of spine circuit from the point A to the coccyx reflex (point B), shown in blue in the S2 Fig a, and the coccyx reflex (point B) is pump 3x (pressure/ release). 4) Starting with the left foot, then the right foot. The emotional circuit is stimulated, starting with the diaphragm reflex (zone 11), 9 turns around the foot, then 3 turns around the foot in front of the diaphragm insertion reflex (zone 12) (see S2 Fig c). The stress axis zones are stimulated simultaneously: adrenal reflex (point 13), epiphysis, hypothalamus, and pituitary (point 14, 15, 16 respectively, see S2 Fig c). 5) Ending with the beginning of spine circuit to coccyx reflex (point B), shown in blue in S2 Fig a, then the coccyx reflex (point B) is pump 3x (pressure/ release). Foot complete smoothing x3, ankle smoothing x3. **SM circuit:** Starting with the left foot, then the right: foot complete smoothing x3, ankle smoothing x3. Starting from the toes to the heel, moving up to the calf, back and circling to the ankle and moving up to the toes. Foot complete smoothing x3, ankle smoothing x3.
(TIF)

**S3 Fig. Connectivity between the left nucleus accumbens and two clusters.** A) Representation of the clusters on a 3D brain. Cluster +02 + 16–10 is formed of 255 voxels with 170 voxels covering 15% of the subcallosal cortex, 9 voxels covering 11% of the right nucleus accumbens and 3 voxels covering a non-labelled area. Size p-FDR = 0.000678, size p-unc = 0.000029. Cluster +02 + 52–26 is formed of 137 voxels, with 83 voxels covering 8% of the frontal medial cortex, 43 voxels covering 1% of the right frontal pole and 1 voxel covering less than 1% of the left frontal pole. Size p-FDR = 0.011661, size p-unc = 0.001014. B) Boxplots of the connectivity between the left nucleus accumbens and cluster +02 + 16–10 according to visits and groups. There was no significant difference at baseline (p-value = 0.14). After intervention there was a significant decrease in the FR group (mean at V1: 0.249 ± 0.094, mean at V5: 0.130 ± 0.101, difference: −0.119 ± 0.097, p < 0.001) and a significant increase in the SM group (mean at V1: 0.156 ± 0.113, mean at V5: 0.287 ± 0.107, difference: 0.131 ± 0.066, p < 0.001). C) Boxplots of the connectivity between the left nucleus accumbens and cluster +02 + 52–26 according to visits and groups. There was a significant difference at baseline of 0.140, SEM: 0.042 (p-value = 0.01). After intervention there was a significant decrease in the FR group (mean at V1: 0.122 ± 0.110, mean at V5: −0.020 ± 0.114, difference: −0.142 ± 0.109, p < 0.001) and a significant increase in the SM group (mean at V1: −0.018 ± 0.116, mean at V5: 0.086 ± 0.104, difference: 0.104 ± 0.145, p-value = 0.002). Blue= Foot Reflexology (FR); light blue= Sham Massage (SM).
(TIFF)

**S4 Fig. Change of functional connectivity in the SMN after intervention.** A) 3D representation of the cluster formed of the left pre and post central gyrus, the right pre and post central gyrus and the supplementary motor area.

p-FDR: 0.039914, p-unc: 0.013305. B) Connectivity of this cluster before and after intervention. Mean connectivity at V1: 0.153 ± 0.227. Mean at V5: 0.018 ± 0.236. Orange = Visit 1 and salmon = Visit 5.
(TIFF)

**S1 Table. Demographic and clinical data at baseline for responders and non-responders.** Means ± standard deviations; MoCA: Montreal Cognitive Assessment; LEDD: levodopa equivalent daily dose; MDS-UPDRS III: Movement Disorder Society – Unified Parkinson's Disease Rating Scale part III; VAS: Visual Analogue Scale; KPPS: King's Parkinson Pain Scale; BPI: Brief Pain Inventory; SF-MPQ: Short Form McGill Pain Questionnaire; CPAQ-8: Chronic Pain Acceptance Questionnaire 8; HAD: Hospital Anxiety and Depression. For all variables, there was no statistical difference between groups p > 0.05.
(DOCX)

**S1 Checklist. CONSORT Checklist.**
(DOC)

**S2 Checklist. CONSORT Extension NPT 2017.**
(DOCX)

**S1 Protocol. Study Protocol** .
(DOCX)

**S1 Data. Data supporting this article.**
(CSV)

## Acknowledgments

We would like to thank all of the patients who participated in this study as well as the Toulouse University Hospital, the Inserm/UPS UMR1214 ToNIC Technical Platform for the MRI acquisitions, Maëlle Huissoud and Bérangère Rabeau who participated in this study and our sponsor.

## Author contributions

**Conceptualization:** Christine Brefel-Courbon, Emeline Descamps.

**Data curation:** Vanessa Rousseau.

**Formal analysis:** Karel Joineau.

**Funding acquisition:** Estelle Harroch, Christine Brefel-Courbon.

**Investigation:** Karel Joineau, Margherita Fabbri, Clémence Leung, Fabienne Ory-Magne, Christine Brefel-Courbon.

**Methodology:** Karel Joineau, Vanessa Rousseau, Patrice Peran.

**Project administration:** Estelle Harroch.

**Software:** Karel Joineau.

**Supervision:** Estelle Harroch, Mathilde Boussac, Patrice Peran, Christine Brefel-Courbon, Emeline Descamps.

**Validation:** Estelle Harroch, Mathilde Boussac, Margherita Fabbri, Clémence Leung, Fabienne Ory-Magne, Vanessa Rousseau, Patrice Peran, Christine Brefel-Courbon, Emeline Descamps.

**Writing – original draft:** Karel Joineau.

**Writing – review & editing:** Estelle Harroch, Mathilde Boussac, Margherita Fabbri, Clémence Leung, Fabienne Ory-Magne, Vanessa Rousseau, Patrice Peran, Christine Brefel-Courbon, Emeline Descamps.

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
