## [Decision Letter · Decision Letter 0]

Dear Dr. Joineau,

Thank you for submitting your manuscript to PLOS ONE. After careful consideration, we feel that it has merit but does not fully meet PLOS ONE’s publication criteria as it currently stands. Therefore, we invite you to submit a revised version of the manuscript that addresses the points raised during the review process.

We look forward to receiving your revised manuscript.

Kind regards,

Xize Jia

Academic Editor

PLOS ONE

Journal Requirements:

1. Please ensure that your manuscript meets PLOS ONE's style requirements, including those for file naming. The PLOS ONE style templates can be found at https://journals.plos.org/plosone/s/file?id=wjVg/PLOSOne_formatting_sample_main_body.pdf and https://journals.plos.org/plosone/s/file?id=ba62/PLOSOne_formatting_sample_title_authors_affiliations.pdf.

2. In the online submission form, you indicated that [The data that support the findings of this study are available in the Supporting Information files. The raw MRI data are available on reasonable request to the corresponding author. The protocol is available on the journal webpage of this article.].

Reviewers' comments:

Reviewer's Responses to Questions

**Comments to the Author**

1. Is the manuscript technically sound, and do the data support the conclusions?

Reviewer #1: Partly

Reviewer #2: Partly

Reviewer #3: Partly

Reviewer #4: Partly

2. Has the statistical analysis been performed appropriately and rigorously?

Reviewer #1: No

Reviewer #2: Yes

Reviewer #3: Yes

Reviewer #4: Yes

3. Have the authors made all data underlying the findings in their manuscript fully available?

Reviewer #1: Yes

Reviewer #2: Yes

Reviewer #3: Yes

Reviewer #4: Yes

4. Is the manuscript presented in an intelligible fashion and written in standard English?

Reviewer #1: Yes

Reviewer #2: Yes

Reviewer #3: Yes

Reviewer #4: Yes

Reviewer #1: It appears that all the comparative analysis in the primary is simple and done done correctly with the correct conclusions being drawn from the data. However, the design and analysis needs a more formal development.

It looks as if the sample size is based on a parametric assumption and some of the analysis is done nonparametrically which makes sense given the small sample. The sample size needs further justification.

For the design what is the rationale for the 20 mm difference and 15 sd. and what reference is being used to make the calculation? There are many analyses done with a limited sample. One should be aware of possible multiple comparison issues.

The secondary and exploratory analysis is interesting and appears to have been well executed, However, some detail is needed for clarification. How was the general linear model implemented? Clustering and multivariate statistics are involved. One reads the Annex and Figure 2 and notes some detail and the clustering. However , what other multivariate statistics are used and how?

Reviewer #2: 1、 The authors investigated the effects of foot reflexology on chronic pain in patient

with Parkinson's disease and associated changes in brain functional connectivity . The methodology mentions that "The dopaminergic medication as well as antalgic treatment had to remain stable for at least 4 weeks before and throughout the study." However, specific details about baseline antalgic treatment (e.g., whether participants were taking antalgic medicine and medication history, types, and dosages) were not provided. Considering the potential impact of antalgic on intervention outcomes, this information should be clarified and analyzed.

2、In the study design, participants received one 60-minute FR or SM intervention every three weeks, totaling four sessions. Could the relatively low frequency of interventions have influenced the effectiveness of the treatment? This possibility should be addressed.

3、The study included participants with various types of pain, but no subgroup analysis based on pain type or stratification by pain severity was performed. This lack of detailed analysis makes the study design appear somewhat simplistic.

4、The baseline data show that the duration of pain symptoms among participants is longer than the duration of their PD. Is this finding reasonable? The authors should provide an explanation for this observation.

Reviewer #3: The current study examined the therapeutic effect of foot reflexology (FR) in Parkinson’s disease (PD) patients with chronic pain using a randomized controlled trial, in which the therapeutic effects were assessed from both behavioral and neuroimaging perspectives. However, the results showed that there was no significant therapeutic effect of FR compared to sham massage (SM). The description of the significance and feasibility of fMRI analysis and ROI selection in the current study needs to be improved. Here are some of my specific comments:

Abstract

1. Authors should reorganize the Abstract section to comply with the formatting requirements of journal.

Introduction

1. The second paragraph of the Introduction should be switched with the third paragraph, and the second paragraph is too simplistic, with insufficient introduction of the significance and feasibility of why the insula, the nucleus accumben, and the mPFC were chosen as ROIs. In addition, the feasibility and significance of ROI-to-ROI analyses are missing.

2. The last two paragraphs of the introduction could be merged.

Methods

1. The present treatment protocol for FR comprises four FR treatments, each with a duration of one hour, administered with a three-week interval between sessions. How was this protocol determined?

2. Why is the pain threshold measured using thermal stimulation?

3. In section of Seed-to-Voxel functional connectivity, the coordinates of the mPFC and the radius of all seed need to be reported.

4. The description of the ROI-to-ROI analysis is not clear, why these networks were chosen is not presented in the Introduction or Methods. And how were these networks obtained, was it through ICA? Finally, did the ROI-to-ROI analysis also use GRF correction?

5. In first paragraph of section of 2.5 Statistical analyses, was it an estimate of the sample size? If so, what is the basis for estimate?

6. The authors seem to have only examined the correlation between FC and VAS, and the authors might also consider examining whether there is a correlation between FC and KPPS or CPAQ-8.

Results

1. Supplementary analyses should be added to the methods section. In addition, it is not clear why analyses were conducted using precentral gyrus, postcentral gyrus and supplementary motor area.

2. In Table 2, ‘*’ appears more appropriately after the P-value.

3. Statistical values that appeared in the manuscript should be italicized, such as p.

Discussion

1. I disagree with the discussion in the third paragraph of Discussion, which categories FR as a form of manual therapy and suggests it is effective. However, the current study showed that there was no significant difference between FR and SM (placebo), which suggested that FR had no actual therapeutic effect.

Reviewer #4: Title: Effect of foot reflexology on chronic pain in Parkinson’s disease: A randomized controlled trial (ID: PONE-D-25-02337)

The authors explored the effectiveness of Foot Reflexology (FR) and Sham Massage (SM) on PD patients with chronic pain. The VAS scores showed no significance between FR group and SM group, while the brain functional connectivity has some difference. This study provides new insights into PD. However, there are some issues that should be classified.

Abstract

1.It is suggested to improve the methods section in the Abstract.

2.More detailed results are better.

Introduction

3.More information on the pathophysiology of pain and studies of rs-fMRI on the brain function alterations in PD would be better.

4.It is recommended to integrated the last two paragraphs of Introduction.

Methods

5.Why didn't this study set up a normal control group?

6.What does adverse events usually mean?

7.In rs-fMRI analysis, why were the insula, the accumbens, and the mPFC chosen as ROI?

8.It would be better to placed the specific coordinate points of mPFC and accumbens.

9.What does NNCP refer to (line241)?

10.For statistical analysis, in my opinion, the first paragraph regarding the sample size was not necessary.

11.Bonferroni corrections method was used for multiple comparisons (line280), however, GRF method was mentioned before (line236), which one was it?

Discussion

12.The discussion regarding the changes of functional connectivity should be deeper.

13.What are the possible reasons why FR was not particularly effective in patients with chronic pain in PD in this study, which differs from the results of previous studies?

**Do you want your identity to be public for this peer review?** For information about this choice, including consent withdrawal, please see our Privacy Policy

Reviewer #1: No

Reviewer #2: No

Reviewer #3: No

Reviewer #4: No

---

## [Author Response · Author response to Decision Letter 1]

7 Apr 2025

We thank the editors and reviewers for their relevant interrogations and comments.

Line numbers refer to the file with tracked changes.

Editors:

1. Please ensure that your manuscript meets PLOS ONE's style requirements, including those for file naming. The PLOS ONE style templates can be found at https://journals.plos.org/plosone/s/file?id=wjVg/PLOSOne_formatting_sample_main_body.pdf and https://journals.plos.org/plosone/s/file?id=ba62/PLOSOne_formatting_sample_title_authors_affiliations.pdf.

As requested, the files names have been modified according to your policies as well as the format of the affiliations and the main body.

2. In the online submission form, you indicated that [The data that support the findings of this study are available in the Supporting Information files. The raw MRI data are available on reasonable request to the corresponding author. The protocol is available on the journal webpage of this article.].

Raw MRI data are considered medical information in Europe. As the protocol did not explicitly stated that the data would be made available, we propose, for ethical or legal reasons, the alternative of providing them upon request. Additionally, we have already shared connectivity values in “S1 Data. Data supporting this article”.

The captions of the supporting information files have been added to the manuscript lines 657-727). We renamed the files accordingly.

Reviewer #1: It appears that all the comparative analysis in the primary is simple and done correctly with the correct conclusions being drawn from the data. However, the design and analysis need a more formal development.

It looks as if the sample size is based on a parametric assumption and some of the analysis is done nonparametrically which makes sense given the small sample. The sample size needs further justification.

For the design what is the rationale for the 20 mm difference and 15 sd. and what reference is being used to make the calculation? There are many analyses done with a limited sample. One should be aware of possible multiple comparison issues.

Due to the particular nature of the data and the limited sample size, we have chosen to process the clinical data with non-parametric assumptions. The connectivity values however were perfectly parametric and processed accordingly. We believe we have limited the number of analyses to the bare minimum considering our objectives.

Based on a previous study in PD patients (Nebe A, Ebersbach G. Pain intensity on and off levodopa in patients with Parkinson’s disease. Mov Disord 2009;24:1233–1237), we assumed an expected difference between reflexology and Sham massage of 20 mm and and a standard deviation of 15.0mm. In addition, a difference from 10 to 20 mm is usually considered as the clinically important difference between analgesic treatment and placebo (Dworkin RH, Turk DC, McDermott MP, et al. Interpreting the clinical importance of group differences in chronic pain clinical trials: IMMPACT recommendations. Pain 2009;146:238–244).

For the design, additional information can be found in Figure 1 of the Supplementary information. Due to the journal’s policy, the design figure could not be loaded as Figure 1. We believe that we have provided all the mandatory elements in the design section according to the CONSORT.

The secondary and exploratory analysis is interesting and appears to have been well executed, However, some detail is needed for clarification. How was the general linear model implemented? Clustering and multivariate statistics are involved. One reads the Annex and Figure 2 and notes some detail and the clustering. However, what other multivariate statistics are used and how?

In Seed-to-Voxel. In the first level analysis, the seed’s activation time course is correlated to the activation time courses of every voxel in the brain. In the second level analysis, a general linear model is automatically computed by the CONN software depending on the study design and the selected statistical contrast (in our case, a 2x2 repeated measures ANOVA). The voxels that are above the statistical threshold are grouped into clusters. The clusters sizes are corrected using the Gaussian Random Field Theory.

In ROI-to-ROI. In the first model analysis, the activation time course of each ROI is correlated to every other by pairs. In the second model analysis, clusters of ROIs are formed using a hierarchical clustering method based on ROI-to-ROI anatomical proximity and functional similarity metrics. Then the Functional Network Connectivity (FNC) method evaluate the within- and between-clusters functional connectivity using a general linear model (Jaffri, 2008).

We processed the second level analyses with the default method in CONN because we had no a priori on the estimated effect size.

The term “multivariate statistics” refers to the statistical contrast implemented in the second level analysis. Here, it is the 2x2 repeated measures ANOVA which belongs to multivariate statistical analyses. In CONN, it is also possible to adjust with covariables, but we have chosen not to adjust our analyses due to the limited sample size and a lack of statistical power.

Reviewer #2: 1、 The authors investigated the effects of foot reflexology on chronic pain in patient with Parkinson's disease and associated changes in brain functional connectivity. The methodology mentions that "The dopaminergic medication as well as antalgic treatment had to remain stable for at least 4 weeks before and throughout the study." However, specific details about baseline antalgic treatment (e.g., whether participants were taking antalgic medicine and medication history, types, and dosages) were not provided. Considering the potential impact of antalgic on intervention outcomes, this information should be clarified and analyzed.

Only few patients were receiving analgesic treatment, and among those who did, it was mostly paracetamol taken irregularly as they deem necessary to relieve their pain. Therefore, we could not include this parameter as covariate neither analyze the dose as a clinical variable. This was added to the limits (lines 492-494).

2、In the study design, participants received one 60-minute FR or SM intervention every three weeks, totaling four sessions. Could the relatively low frequency of interventions have influenced the effectiveness of the treatment? This possibility should be addressed.

The specific reflexology protocol used in the present study was developed in collaboration with experienced reflexology practitioners. It reflects the reality of reflexology practice, considering the body's response, which typically occurs within three weeks for chronic conditions (Faure Anderson, 2018, EAN. 9782813217707). However, the peak effect varies between individuals and remains uncertain. The protocol was validated by the working group on reflexology evaluation (Descamps et al, 2025, DOI : 10.3917/heg.114.0356).

In the context of chronic conditions, the literature generally supports a frequency of one session every three weeks to one month, suggesting that our intervention frequency aligns with current practice. However, regarding the optimal timing for outcome measurement, we had no strong prior assumptions. We therefore selected a three-week interval, consistent with a chronic care framework.

3、The study included participants with various types of pain, but no subgroup analysis based on pain type or stratification by pain severity was performed. This lack of detailed analysis makes the study design appear somewhat simplistic.

We agree that the interventions could not have the same effects depending of the type of pain. We did not have a prior assumption in which type of pain could be better ameliorated by FR. That is why we included all types of pain in PD. We did not process subgroup analyses because of our limited sample size (n=15 in each group) and a lack of statistical power due to the heterogeneity of the data, but we agree that it would be important to address this in a future study.

4、The baseline data show that the duration of pain symptoms among participants is longer than the duration of their PD. Is this finding reasonable? The authors should provide an explanation for this observation.

Pain could be a prodromal feature of Parkinson’s disease and can appears years before the motor symptoms leading to the diagnosis (Buhman, 2020, doi: 10.1007/s00415-017-8426-y), so this finding is coherent with clinical and in research observations.

Reviewer #3: The current study examined the therapeutic effect of foot reflexology (FR) in Parkinson’s disease (PD) patients with chronic pain using a randomized controlled trial, in which the therapeutic effects were assessed from both behavioral and neuroimaging perspectives. However, the results showed that there was no significant therapeutic effect of FR compared to sham massage (SM). The description of the significance and feasibility of fMRI analysis and ROI selection in the current study needs to be improved. Here are some of my specific comments:

Abstract

1. Authors should reorganize the Abstract section to comply with the formatting requirements of journal.

Plos One requests to comply with the CONSORT checklist for an RCT. For the abstract, we believed that we provided all necessary items in that checklist, and we added some details in the methods and results.

Introduction

1. The second paragraph of the Introduction should be switched with the third paragraph, and the second paragraph is too simplistic, with insufficient introduction of the significance and feasibility of why the insula, the nucleus accumbens, and the mPFC were chosen as ROIs. In addition, the feasibility and significance of ROI-to-ROI analyses are missing.

The second paragraph of the introduction have been switched with the third paragraph.

For the Seed-to-Voxels analyses, these 3 areas were chosen as seeds because they have shown altered connectivity in pain conditions (Tu et al., 2019; Baliki et al., 2006, 2012) and in PD pain (Polli et al., 2016; Kinugawa et al., 2022, Joineau et al., 2024). Studies focusing on chronic pain in PD and using functional connectivity as outcome are very limited.

The main and the most studied networks implicated in these processes are the default mode network (DMN), the ECN and the Salience Network (SN) (Shirer et al., 2012; Richiardi et al., 2015).

The sensory motor network (SMN) is also implicated in attentional processes, involved in tactile touch and pain (Shirer et al., 2012; Richiardi et al., 2015).

Finally, we wanted to explore a new network specifically implicated in pain, the neural network correlates of pain (NNCP) from Descamps et al., 2023.

More details about the selection of seeds and networks were given in the introduction (lines 96-104) and method sections (lines 269-272).

2. The last two paragraphs of the introduction could be merged.

The last two paragraphs regarding the objectives were merged.

Methods

1. The present treatment protocol for FR comprises four FR treatments, each with a duration of one hour, administered with a three-week interval between sessions. How was this protocol determined?

The specific reflexology protocol used in the present study was developed in collaboration with experienced reflexology practitioners. It reflects the reality of reflexology practice and has been subsequently validated by this reflexology evaluation working group (Descamps et al, GER: a Think and Do Tank for the assessment of reflexology, HEGEL, Vol. 15 N°1 - 2025).

2. Why is the pain threshold measured using thermal stimulation?

The nociceptive pain threshold was measured using thermal stimulation because it is one of the common methods to assess this parameter, alongside cold, pressure, and electrical stimulations. Parkinson’s disease patients have a lower nociceptive heat threshold compared to healthy subjects (reviewed by Sung et al., 2017, doi: 10.1016/j.parkreldis.2017.12.031). We expected that FR could normalize it which was not the case.

3. In section of Seed-to-Voxel functional connectivity, the coordinates of the mPFC and the radius of all seed need to be reported.

The coordinates of all the seeds have been added as well as the radius size of the insulae and mPFC (lines 256-262). For the nuclei accumbens, the segmentation was implemented in the CONN software and do not have the shape of a sphere.

We have chosen not to use the default atlas for the insulae and the mPFC as we wanted to separate the anterior and posterior parts of the insulae and because the mPFC is not well defined in the default Harvard-Oxford atlas directly implemented in CONN.

4. The description of the ROI-to-ROI analysis is not clear, why these networks were chosen is not presented in the Introduction or Methods. And how were these networks obtained, was it through ICA? Finally, did the ROI-to-ROI analysis also use GRF correction?

We thank the reviewer for this question which provides an opportunity to further clarify the criteria for networks selection.

In the introduction, we have added a paragraph showing that PD patients have an alteration of networks associated with attentional processes such as the executive control network (ECN) (Tan et al., 2015) (lines 94-106).

The main and the most studied networks implicated in these processes are the default mode network (DMN), the ECN and the Salience Network (SN) (Shirer et al., 2012; Richiardi et al., 2015).

The sensory motor network (SMN) is also implicated in attentional processes, involved in tactile touch and pain (Shirer et al., 2012; Richiardi et al., 2015).

Finally, we wanted to explore a new network specifically implicated in pain, the neural network correlates of pain (NNCP) from Descamps et al., 2023.

The DMN, SN, ECN and SMN were directly obtained from atlas files of Shirer et al., 2012; Richiardi et al., 2015. This atlas is often called the “Willard” atlas or the “Standford atlas”.

The NNCP was obtained from Descamps et al., 2023.

We did not perform any ICA or other analyses to obtain the networks, we used their respective atlas files.

5. In first paragraph of section of 2.5 Statistical analyses, was it an estimate of the sample size? If so, what is the basis for estimate?

The first paragraph of this section is indeed an estimation of the sample size needed for this study and with the VAS as first outcome.

Based on a previous study in PD patients (Nebe A, Ebersbach G. Pain intensity on and off levodopa in patients with Parkinson’s disease. Mov Disord 2009;24:1233–1237), we assumed an expected difference between reflexology and Sham massage of 20 mm and and a standard deviation of 15.0mm. In addition, a difference from 10 to 20 mm is usually considered as the clinically important difference between analgesic treatment and placebo (Dworkin RH, Turk DC, McDermott MP, et al. Interpreting the clinical importance of group differences in chronic pain clinical trials: IMMPACT recommendations. Pain 2009;146:238–244).

6. The authors seem to have only examined the correlation between FC and VAS, and the authors might also consider examining whether there is a correlation between FC and KPPS or CPAQ-8.

Since the main outcome of the study was the VAS, we decided to correlate this main outcome with the exploratory one: funct

---

## [Decision Letter · Decision Letter 1]

Effect of foot reflexology on chronic pain in Parkinson’s disease: A randomized controlled trial

PONE-D-25-02337R1

Dear Dr. Joineau,

We’re pleased to inform you that your manuscript has been judged scientifically suitable for publication and will be formally accepted for publication once it meets all outstanding technical requirements.

Kind regards,

Federico Giove, PhD

Academic Editor

PLOS ONE

Additional Editor Comments (optional):

Reviewers' comments:

Reviewer's Responses to Questions

**Comments to the Author**

Reviewer #1: All comments have been addressed

Reviewer #2: (No Response)

Reviewer #3: All comments have been addressed

Reviewer #4: All comments have been addressed

2. Is the manuscript technically sound, and do the data support the conclusions?

Reviewer #1: (No Response)

Reviewer #2: (No Response)

Reviewer #3: Yes

Reviewer #4: Yes

3. Has the statistical analysis been performed appropriately and rigorously?

Reviewer #1: (No Response)

Reviewer #2: (No Response)

Reviewer #3: Yes

Reviewer #4: Yes

4. Have the authors made all data underlying the findings in their manuscript fully available?

Reviewer #1: (No Response)

Reviewer #2: (No Response)

Reviewer #3: Yes

Reviewer #4: Yes

5. Is the manuscript presented in an intelligible fashion and written in standard English?

Reviewer #1: (No Response)

Reviewer #2: (No Response)

Reviewer #3: Yes

Reviewer #4: Yes

Reviewer #1: (No Response)

Reviewer #2: (No Response)

Reviewer #3: (No Response)

Reviewer #4: (No Response)

**Do you want your identity to be public for this peer review?** For information about this choice, including consent withdrawal, please see our Privacy Policy

Reviewer #1: No

Reviewer #2: **Yes: ** Xiaohong Li

Reviewer #3: No

Reviewer #4: No
